# An Extended AI-Experience: Industry 5.0 in Creative Product Innovation

**DOI:** 10.3390/s23063009

**Published:** 2023-03-10

**Authors:** Amy Grech, Jörn Mehnen, Andrew Wodehouse

**Affiliations:** Department of Design, Manufacturing and Engineering Management, University of Strathclyde, Glasgow G1 1XJ, UK

**Keywords:** virtual reality, artificial intelligence, creativity, ideation, brainstorming

## Abstract

Creativity plays a significant role in competitive product ideation. With the increasing emergence of Virtual Reality (VR) and Artificial Intelligence (AI) technologies, the link between such technologies and product ideation is explored in this research to assist and augment creative scenarios in the engineering field. A bibliographic analysis is performed to review relevant fields and their relationships. This is followed by a review of current challenges in group ideation and state-of-the-art technologies with the aim of addressing them in this study. This knowledge is applied to the transformation of current ideation scenarios into a virtual environment using AI. The aim is to augment designers’ creative experiences, a core value of Industry 5.0 that focuses on human-centricity, social and ecological benefits. For the first time, this research reclaims brainstorming as a challenging and inspiring activity where participants are fully engaged through a combination of AI and VR technologies. This activity is enhanced through three key areas: facilitation, stimulation, and immersion. These areas are integrated through intelligent team moderation, enhanced communication techniques, and access to multi-sensory stimuli during the collaborative creative process, therefore providing a platform for future research into Industry 5.0 and smart product development.

## 1. Introduction

In today’s corporate world, ideas and problems are becoming increasingly complex [1]. Whilst group ideation techniques aimed at generating creative solutions such as ‘brainstorming’ are highly useful, they are often used in situations where these rules are not emphasised or followed closely [2]. Furthermore, there are several challenges and limitations in group ideation which led to idea fixation or rejection [3]. Additionally, to solve complex problems, teams having diverse competences are required to collectively find innovative solutions [4], which has led to the utilisation of digital tools such as videoconferencing. However, the artificial environment in remote meetings can lead to a lack of focus which could negatively impact ideation [5]. Creativity is one of the main driving forces behind engineering design and is therefore key within business environments [6], which highlights the need for current challenges related to group ideation and problem complexity to be addressed. Research [7,8,9] has demonstrated the potential of brainstorming activities occurring with the support of Virtual Reality (VR) and Artificial Intelligence (AI) technologies due to their immersive elements and ability to process complex tasks. However, the integration of both VR and AI technologies has not been extensively explored, highlighting a research gap. Therefore, the main research question is to explore how the combination of VR and AI can augment collaborative and creative ideation through a human-centric approach, using technological capabilities offered by VR and AI, as described in Figure 1. By merging the strengths of both technologies, a design ideation environment is generated where participants are guided through the brainstorming process via exposure to the environment, information, and artefacts that would not be achievable in a conventional setting. This is achieved by combining immersive elements offered by VR with AI fields such as information extraction. The result would involve merging of human and technical strengths to produce a beneficial outcome for both. This is a core concept of Industry 5.0 whereby the creativity skills of humans are to be further enhanced through the support of technology aimed for product innovation [10]. The outcome of this research entails a novel platform for future empirical research aimed at allowing brainstorming activities to be reclaimed as a truly challenging and inspiring activity where participants are fully engaged in AI-supported VR environments. 

## 2. Research Methodology

This section discusses the methodology applied in this literature review. Following the identification of the research question, the main keywords were identified. The main themes related to this literature search were “virtual reality”, “artificial intelligence” and “creativity”. Alternative terminology was considered including broader and narrower terms, synonyms, and antonyms. The selected main databases were Google Scholar^®^ (Mountain View, CA, USA) and SCOPUS^®^ (Amsterdam, The Netherlands) for academic related publications. A preliminary search was conducted, whilst iteratively identifying any appropriate exclusion criteria based on the results obtained. The search included documents published since 2012 to consider research in line with recent levels of technology and written in the English language. The result revealed 244 results. Once the final refined search was performed and key papers were identified, they were reviewed based on the main outcomes, strengths, weaknesses, and the methodology applied. The snowball technique was applied to identify additional key papers and authors in the field. 

To determine the state-of-the-art in the field, a bibliographic analysis (Figure 2) was performed linking all three main fields explored in this research which was based on the final refined search performed using SCOPUS^®^. From the analysis, “virtual reality” and “artificial intelligence” terminologies are dominant as evident by the size of the green keyword cluster surrounding both terminologies located centrally in Figure 2. The bibliographic analysis was analysed according to the main groups of keywords which are categorised by their colour. The green keyword group mainly concentrates on keyword clusters or terms linked with VR such as gaming and human–computer interaction. AI-related terms are also highly linked with the green cluster as denoted by the terms “artificial intelligence” and “ambient intelligence”. The green cluster contains a multitude of connections with the purple and yellow clusters which are in proximity. This group delves further into applications and terms that involve the specific use of AI such as “e-learning”, “deep learning”, “Internet of Things” and “Industry 4.0”. The education sector is highly linked with such keywords demonstrating the close link between AI- and VR-based technologies in the field of education, as shown by the yellow keyword group located in between the purple and green keyword groups. Furthermore, applications in design and manufacturing, which this proposal directly addresses, are not strongly present in the analysis.

Creativity-related terms are in the red keyword group. This group contains keywords of smaller sized clusters which demonstrates how creativity-related terms occur less frequently in literature when applied in combination with VR and AI related terms. The lack of creativity-related terminology demonstrates limited interconnectivity of creative concepts with VR and AI. This further confirms the current opportunity in linking creativity in design to AI and VR related technologies in this research. The area of opportunity, which this research aims to address, is labelled in Figure 2. The term “conceptual design” links “creativity” with “AI”, denoted by its proximity with both terms and is centrally located in Figure 2. A similar scenario is present in the blue keyword group in which the centrally located term “machine learning” strongly links “AI” with the human involvement related to the search.

### State-of-the-Art VR and AI Brainstorming Applications

In exploring some of the most relevant work inside the area of opportunity highlighted in Figure 2, there were efforts to provide technological support for brainstorming via VR and AI, though not of the integrated nature set out in this article. For example, Perlin, He, and Zhu [11] have transformed the concept of freeform sketching used in brainstorming sessions happening in person into a virtual environment. Free-form sketches are translated and interpreted by the system using an AI recognition system in real time. This system can also convert ideas drawn in mid-air by participants into elements that allow for building complex simulations with the aim of enhancing creative discussion. In their later study [12], they have leveraged this application into a system that converts 2D sketches into 3D models and generates real-time procedural animations. Results reveal positive influences in communication between participants that fosters collaboration. Alternatively, Lee, Feldman, and Fu [13] focused on mind mapping to create concepts through visualisation, data associations and patterns. AI was considered as a solution to generate real-time automated mind maps from verbal communication using a speech-to-text algorithm and keyword processing. The study assessed the accuracy and reliability of this system compared to human-generated mind maps. The results conducted in this study conclude that the generated mind maps were 80% similar to those that are human generated. This demonstrates the algorithm’s capability of augmenting manually generated mind maps which are prone to productivity loss due to the conversational flow and documentation processes, through the synchronous capture of ideas. In future, the effectiveness and applicability of the algorithm is expected to be evaluated for real-time mind mapping whilst further developing the keyword extraction and relations process. 

Another important element of ideation is communication through body language. The experiment by Ide et al. [14] aimed to understand the impact of symbolic gestures to express pre-determined intentions and emotions and therefore, enhance communication amongst team members through avatars. A limitation of this study was the learning process involved for members to apply specific gestures; however further research is expected. The results indicated that the symbolic body language enhanced social presence and encouraged open discussion in VR, which is a vital element in brainstorming activities. 

An aspect that was not considered in the above-mentioned studies was the role of the moderator in a brainstorming environment. The moderator motivates discussion and intervenes when brainstorming rules are not abided [15]. Research by Strohmann, Siemon, and Robra-Bissantz [16] used AI technologies such as machine learning, reasoning, and natural language processing to successfully generate an intelligent moderator. An intelligent moderator was also developed by Wang, Cosley and Fussell [17] which was responsible for generating pictorial stimuli and subsequently also generated successful creative output. These studies [16,17] demonstrate the potential of AI to facilitate to role of the moderator in the brainstorming process; however, they did not involve fully immersive virtual environments. Therefore, this research aims to synthesise such findings to generate a virtual environment that merges VR and AI technologies to generate creative output.

To understand the practical implementation of the VR environment, a review of current commercial applications that support design ideation was performed. Most of the tools support group ideation through the generation of mind-maps and free form sketching whilst also providing access to tools that can further support ideation such as whiteboards, presentations, files, and 3D models. Several AI technologies were implemented including speech-to-text functionality, AI-based facial animation, lip-syncing, and eye contact. None of the tools provide the role of a virtual moderator which this research aims to explore. Persistent content was also generally available to allow members to access their previous work at any time. Table 1 summarises a non-exhaustive list of the tools reviewed and includes descriptions of their key features.

## 3. Creativity in Engineering Design Teams

Amabile [18] defines creativity as the production of ideas that are novel. Creativity can be associated with ideas that develop from existing solutions. In a relatively stable environment, it is more likely that incremental ideas would be generated from existing competencies as opposed to divergent or ground-breaking ideas. To manage creativity within a company, one must therefore comprehend the conditions under which people tend to express ideas, whether they are convergent or divergent [19].

In an engineering design context, creativity involves generating ideas and solving design problems that render a successful and competitive product or service [5]. Whyte et al. [6] claim that creativity is one of the main driving forces behind engineering design and is strategic within the business environment. Idea generation is often a group activity that draws on the variety of knowledge and skillsets in multi-disciplinary design engineering teams. Common logic suggests that team ideation combines these different perspectives to produce a richer set of overall results—‘two heads are better than one’. While several studies have shown that group ideation results in lower productivity than individual working (i.e., fewer ideas generated per participant), others argue that there are additional benefits to interactive groups, which include improving organisational memory, building and pooling of designer knowledge and skills, creating an attitude of support, and facilitating the building of ideas to create novel solutions [3,20,21].

### Current Ideation Techniques

There are several creativity-enhancing techniques or ideation methods that can be applied during ideation by design and business academia, industry, and creativity specialists. The most common and perhaps the simplest, is the classic brainstorming method, first described by Osborn in 1953 [22], in which a group of people generate ideas together (verbal and/or written or sketched) while following a set of rules, listed as follows: (1) aim for quantity; (2) avoid criticism; (3) build on ideas; (4) wild ideas are welcome.

However, due to its popularity the term ‘brainstorming’ is often used in situations where these rules are not emphasised or followed closely [2]. Furthermore, there are several challenges and limitations to this largely unstructured approach to group ideation. For example, the group may limit their solution space due to self-censorship, premature rejection of ideas or idea fixation [3]. Brainstorming can also lead to unequal input from all participants, either due to one or more participants dominating the discussion or ‘social loafing’, where participants lower their input due to reduced personal responsibility [2,22]. Other limitations in group ideation when compared to that done in an individual level are related to group dynamics, evaluation apprehension in which participants are reluctant to express their ideas due to fear of negative appraisals, and production blocking [23] which occurs when individuals take turns in expressing ideas. Production blocking is mainly caused by cognitive interferences that occur between participants’ generating and articulating this idea, which could lead to a loss in productivity [24]. Such limitations can also lead to ‘paradigm preserving’ approaches to ideation which do not encourage participants to adequately challenge their perspectives on the design task to arrive at novel and surprising solutions [25]. Garfield et al. [26] and Dennis, Minas and Bhagwatwar [23] claimed that paradigm-modifying ideas from one participant, therefore exposing the team to ideas that more drastic compared to existing solutions, would positively influence the rest of the group via cognitive priming. Another factor that can impact the quality level of ideation is the preferred medium preference for ideation of individuals, whether they choose to express ideas individually or in a group or whether they opt for sketching or verbal ideation. Individuals with better sketching skills can lead to more efficient interpretation of ideas to others [27]. This could potentially be a limiting factor for people with lower sketching skills to translate ideas effectively, particularly those related to complex design solutions. 

Traditionally, design teams were collocated and consisted of face-to-face interactions. In today’s corporate world, ideas and problems are becoming increasingly complex [1]. This has led to people having diverse competences, work together to find innovative solutions [4]. This has been facilitated through the utilisation of electronic communications such as videoconferencing, where teams can collaborate without being in the same physical environment [1]. The artificial environment in remote meetings can lead to a lack of concentration on creativity since it is easier for members to divert their attention elsewhere. This is enhanced when there are visibility barriers and technological issues such as poor sound quality [5]. 

A critical area with regard to creativity in specific technical domains is information support. The design consultancy IDEO have for many years utilised something known internally as the “Tech Box” [28] where participants could physically store and refer to existing design exemplars as a tool to assist concept development. The physical contents could be handled, discussed, discarded and even catalogued amongst team members [29]. Smith, Kohn and Shah [30] conducted studies which concluded that the type of material presented to the designer has an influence on the output of a concept design session: seeing commonplace ideas resulted in unoriginal designs, but seeing novel ideas resulted in more unique designs. The type of information can also affect subsequent output. Benami and Jin’s [31] study on cognitive processes resulted in a recommendation that stimuli should be “meaningful, relevant, and ambiguous” for optimal design performance. Ambiguity may lend itself well to blue-sky thinking but as discussed above, most of the conceptual design work occurs within a fixed set of design parameters, and whatever resources are provided should take account of these—for example, reference to competitor products is not at all ambiguous but can be very useful to increase awareness of alternative approaches and to clarify where potential gaps in a market may lie. There should also be an awareness of how knowledge of a particular problem can inhibit as well as enhance creativity [32]. A certain level of knowledge is necessary to contribute meaningful ideas, but over-familiarity with a problem or field can lead to fixation and an inability to develop solutions from alternative perspectives [33].

Additionally, teams having a centred leadership, meaning someone that supports the creative process from beginning to end was also found to contribute positively to the creative process [5]. For a brainstorming session to be effective, it requires the role of a facilitator or moderator [16] who motivates discussion and intervenes when brainstorming rules are not abided [15].

The above techniques and methods have traditionally been used to provide structure to ideation and motivate members to generate ideas. However, there lies the question of how the above ideation techniques can be transformed to a virtual environment to not only replace but enhance creativity with the support of cutting-edge technologies. The following sections introduce VR and AI technologies to help identify in what ways such emerging technologies can be implemented to augment the creative experience. This is achieved in a way that is human-centric and that empowers every individual of a creative and diverse team, hence aligning with values encompassed in Industry 5.0 [10].

## 4. Integrating Virtual Reality (VR) and Creativity

This section discusses a high-level understanding of VR technology and its working principle. VR entails the user to be “immersed” into a 3D environment derived from a computer-generated simulation [34]. The VR environment is accessed through a Head Mounted Display (HMD) that also tracks the position and orientation of the head. In addition, VR requires the use of two hand-held controllers for navigation and haptic feedback [35]. 

Immersion and presence are two characteristics often used to characterise the VR experience. Slater and Wilbur [36] define immersion as the degree to which VR stimulates the sensory receptors of users. It engages the user who needs to perceive and interpret the environment. The greater the immersion, the higher the experience of presence [36]. Felton and Jackson [37] define presence as the degree to which something appears to exist in the same environment as the observer. Humanity’s reviving interest in presence, brought by the interest in VR, is indicative of the human intrinsic ability to perceive VR as an extension of their own senses and experiences [38]. The term social presence is associated with the emotional and cognitive relationship that is developed through interactions between humans and the virtual environment [39]. Therefore, this process makes VR highly beneficial for members wanting to immerse into the virtual environment and interact with it.

A break in presence can occur when the user interacts with the real world due to the virtual environment breaking down for example due to loss of tracking. When the user is unable to see their own body inside the virtual environment, they might also feel disembodied, hence requiring a virtual representation of the user’s body. In addition, VR is subject to sensory incongruences leading to possible adverse health effects during usage such as motion sickness and latency [35].

Although current virtual environments mostly involve visual and auditory senses, further technological advancements involve tactility in which the user experiences physical interaction with the texture and volume of virtual objects and therefore enhances the level of immersion. Temperature sensations and weather conditions can also be felt via costumes [40]. Haptic feedback can be incorporated in tactility, hence simulating forces such as hardness, inertia and weight [41], which can be achieved via haptic gloves. Interactions with the virtual environment could also be done with the real environment which is enabled through augmented reality (AR) and mixed reality (MR) experiences, in which digital information is superimposed in existing surroundings [34]. The terms VR, AR and MR also fall under the umbrella term extended reality (XR) that broadly covers all three [42]. 

### 4.1. Elements Comprising Virtual Brainstorming

Gong et al. [9] merge brainstorming and VR by defining the term virtual brainstorming as a “virtual environment that facilitates the brainstorming process.” It comprises a 3D virtual environment, avatars that represent the members of the team, and brainstorming. Research by Gong et al. [9], Gong and Georgiev [43], and Graesslar and Taplick [44] has analysed how creativity in VR can be enhanced. The findings were synthesised into the main elements that affect virtual brainstorming, as described in Figure 3. The main elements of a virtual brainstorming environment would be the environment itself, the user and the method of communication or interaction applied between the user and that environment or between multiple users. Each element is linked with several factors which are also listed is Figure 3.

The first element (the environment) can support the brainstorming process in multiple ways. The environment contains multiple virtual entities that could encourage design thinking. Presenting mental stimulations can also influence users’ cognitions and behaviours [23]. For instance, positive emotions can be induced through the presence of plants, windows and colours. Fun atmospheres may generate better acceptance by members and motivation to interact with the environment [45,46]. Studies [45,46] explored how virtual objects related to the topic being brainstormed were highly effective at enhancing creativity for creative writing and brainstorming applications. The interaction between users and objects allows users to evaluate and identify the best ideas and find ways to improve on the previous design iteration. This is described as appraisal by Gong et al., [9], who identify this effect as a means of enhancing the quality of the design. Hwang et al. [47] claim that there could be a positive correlation between realism and creativity; therefore, realistic interactions would support product ideation. Multi-sensory stimuli involving virtual, tactile, auditory, and olfactory stimuli within the environment could also support the creative process through the positive effect of immersion on users [44]. The greater the immersion experienced by members, the less distracted members feel which could lead to greater focus and quality ideation [48,49]. 

The second element (users) plays a crucial role in virtual brainstorming environments. Each user is digitally represented as an avatar or a 3D body [9]. Avatars can interact with other avatars and perform actions [50]. Their behaviour is controlled by the human representing it [9]. They also have an influence on human cognition, through the effects of presence, the proteus effect and the social identification effect. Presence empowers the behaviour of users because they would assimilate it to that in a real scenario [51]. The Proteus effect, introduced by Yee and Bailenson [52], explains the behaviour of the user which can be affected by how they are embodied in the virtual environment. The Proteus effect is generated when the user’s digital representation influences their behaviours by creating a behavioural modulation based on the avatar’s appearance. Users can customise avatars to influence their emotions in a positive way and encourage creativity [7]. Social identity effects are related to characteristics that are common and identifiable amongst team members [53]. In the study performed by Bonnardel, Forens and Lefevre [54] and Guegan et al. [7], avatars containing individual identity cues and social identity cues enhanced creativity amongst engineering students. When introducing avatars that were associated with inventors, engineering students produced a higher creative output in terms of novelty and fluency. When avatars wore university suits, social identity was higher, leading to greater creative output. However, a later study by Buisine and Guegan [53] concluded that social identity cues could inhibit creativity. Therefore, more research is required that also includes other disciplines and users. Additionally, Guegan et al. [8] claim that using avatars to maintain anonymity of members could encourage discussion and expression beyond social boundaries, and therefore produce more ideas.

The third element (communication) is vital in virtual environments. If users can choose the communication method that is most appropriate for them, this could motivate members to reach their creativity target [44]. Tracing, which refers to the real-time capturing of the location and movements of the person within the environment, encourages further expressiveness and enhances communication [48,49]. Besides body movements, oral communication has been effective in generating novel ideas [55]. However, further research is required to further enhance communication and comprehension between members such as the effects of eye contact and facial expressions [56]. Voice recognition technology can convert speech to text and therefore provides the possibility of recording ideas and supporting the creative process [48,49]. In addition, automatic recording of the product or process provides members with access to previous design solutions that can assist in obtaining future improvements [7]. In the study performed by Gonçalves and Campos [57], they analysed the effect of adding user interfaces to the environment as means of interaction between the user and the environment, termed as the “mild place illusion”. This was compared with not having an interface at all, termed as the “full-level place illusion”. The result showed a positive increase in self-perceived creativity with the presence of an interface. This is unexpected since fully immersive environments suggest enhanced creativity [48,49,55], hence encouraging more research on the role of interfaces within virtual brainstorming environments.

The above elements and factors are highly valuable when designing and developing environments that foster creativity that is fully integrative to the individual and are influential in this research. In addition, these elements were addressed to further enhance creativity in VR through the support of AI technologies. The following section introduces AI and describes its emerging role in virtual environments. 

## 5. Integrating Artificial Intelligence (AI) and Creativity

Russell and Norvig [58] describe AI as the replication of human abilities including speech utilisation and creativity. This is achieved through algorithms performed by computing machines. Wienrich, and Latoschik [59] extend this definition by using computational models to solve intelligent tasks which humans are unable to perform either because of the quantity and complexity of the input data involved in for instance data mining and big data. The term Human–AI Interaction focuses on the interactions between AI technologies and humans [59]. The relationship between humans and AI technologies is further investigated in this research to analyse how AI can play a role in a brainstorming process which replicates and augments what happens in a face-to-face scenario. 

Recent technological advancements in big data accessibility and computing infrastructures enabled AI to facilitate industrial informatics [60], which involves the integration of information technology techniques in information retrieval and analysis to support industrial applications [61]. The field of information extraction is important in recognising and structuring information from pre-specified sources such as search engines. The process applied in extraction information stems from AI fields including machine learning, logic and search algorithms, pattern recognition, computational linguistics [62] and natural language processing (NLP) [63]. 

Humans interpret information and express ideas through various communication techniques. An essential tool of communication and information exchange is the human voice [64] and therefore technological advancements in the field of speech recognition have become prevalent through the application of AI [65]. Microsoft and Google have also provided cloud speech-to-text capabilities via deep learning models [66], which involve large neural network models that are able to make decisions that are data-driven [67]. Although the accuracy of speech that is read from a text is high, spontaneous speech is more challenging due to its varied structure compared to written text and it is generally made up of fragments of speech that are considered redundant [68]. Therefore, automatic speech summarisation techniques offer the opportunity to extract vital information whilst removing incorrect data [68] which could prove to be a potential solution when transcribing the spontaneous nature of spoken language involved in brainstorming and creative activities.

Besides voice, another form of human interaction is based on movement. AI techniques such as machine learning have increasingly become ubiquitous in human–computer interactions (HCI). Features of machine learning include object detections [69] and pose estimations [70] that are being applied in VR frameworks to create HCI [71]. Simple interactions such as object grabbing are relatively simple to map onto a virtual environment; however, more complex interactions are not as straightforward. Therefore, Interactive Machine Learning (IML) offers an opportunity for developers to implement complex movements in immersive environments via tracking of their own body movements, hence mirroring realistic interactions [72]. The technique is facilitated through artificial neural networks that allow software to capture data based on observations. Implementing movement interactions in VR has proven to be highly beneficial to enhance immersion and presence [51], emotional engagement [31] and improve focus [44]. This proves the potential of such technology for brainstorming scenarios that involves multiple interactions between team members and between humans and virtual elements, which motivates the inclusion of IML for this research. 

As described in Section 4.1, humans are digitally represented as avatars in VR [9]. The moderator used in brainstorming teams could also be represented by a virtual agent. Studying interactions between humans presents challenges due to the difficulty in replicating scenarios and interaction partners. Virtual humans offer standardisation and ecological validity [53], which demonstrates the potential of AI and VR technologies to investigate human interactions. The digital human is created from the notion of a digital twin, which is based on the representation of a real human. When combined with AI, a digital human is capable of interacting with humans through verbal and non-verbal forms of communication. Therefore, intimacy can be developed from this two-way communication that creates an emotional connection with humans [39]. DiPaola and Yalçin [73] addressed the complexity behind building natural conversations between virtual humans through an AI-based deep learning approach which involved human related inputs such as facial emotion recognition, eye gaze, and voice stress to create an AI avatar system that allows empathetic conversations between virtual agents and users [73]. 

The integration of AI with VR environments, also denoted by the term Intelligent Virtual Environments [74], presents other benefits for such applications, which include the possibility of having a design space which embodies interaction partners having varied forms and appearances; AI systems are easily adaptable to specific or elusive user groups and AI offers a high degree of versatility to simulate various tasks and domains [64]. AI also enables multimodal interaction that is based on multiple forms of communication, both verbal and non-verbal. In the research by Lv et al. [75], a combination of intelligent HCI and deep learning approaches were successful in obtaining results of high accuracy, robustness and realism in applications involving emotion recognition, facial recognition and gesture recognition. The next section explores how AI can further augment environments dedicated to brainstorming applications and presents a proposal of an AI- and VR-based environment for this research.

## 6. Discussion

### 6.1. Application of AI and VR for Design Ideation

The configuration of the virtual human-centric brainstorming environment represented by this research is described in Figure 4. In reflecting upon the literature and state-of-the-art, three distinct areas of intervention were identified to support an AI-supported, VR-based ideation activity: facilitation, stimulation, and immersion. 

Facilitation is imparted through the presence of an intelligent moderator who facilitates the discussion in a way that is natural and unbiased [15]. Therefore, a healthy discussion is encouraged that is irrespective of prior skills and status of the participatory members, which preserves the importance of every team member—a core value of Industry 5.0 [10]. Unbiased conversations can happen not only between the facilitator and the participant but also between participants through the preservation of everyone’s anonymity, if the team deems this as valuable in the discussion [8]. Creative stimulation is attributed to the positive atmosphere generated by the colourful environment, plants, and windows [45,46]. Visual mental stimulations are placed on the wall through information extraction to allow ease of visualisation of information [62]. Stimulation is also provided through the access of previously recorded and organised information. This is achieved through voice recognition techniques [48,49], speech-to-text algorithms [13], and natural language processing [63]. Access to the design assets, placed centrally in the virtual space, enables interaction with virtual objects using multiple senses whilst allowing access to previous design iterations. The interaction with 3D assets and with the environment also has a positive influence on the level of immersion of participants [7,9]. Immersion is also influenced by body movement tracking [72] and location tracing of team members [48,49], due to the positive influence on communication. The visual representation of users through avatars facilitates body language and oral communication [50]. These areas will harness AI and VR capabilities to transform characteristics of brainstorming into a digital ideation space.

From this configuration, designers feel empowered to enhance their creative skills through novel uses of digital technology. This human-centric approach presents a framework for industries that promotes competitive product ideation and innovation by combining the strengths of both humans and current technologies. This configuration also provides a pathway towards sustainability and resilience using digital technology when compared to the usage of physical environments and prototypes. This transformative model aligns with the vision of Industry 5.0 that focuses on human growth and well-being through alternative modes of technology—in this case, an AI- and VR-based environment for enhancing creativity within design engineering [76]. The nature and routes to implementation are discussed further in the next section.

#### 6.1.1. Key Area of Intervention: Facilitation

To mitigate any challenges and support creative ideation activities, brainstorming requires a skilled facilitator to function effectively and too often there is nobody willing or able to assume this role. It is perceived as less ‘fun’ than simply generating ideas; it requires concentration and knowledge of best practice, as well as a degree of social and emotional literacy to maintain a positive attitude and pace to discussions [15]. There is an opportunity to address this using an AI agent that is not only familiar with the rules of brainstorming but can understand the group dynamics at play at a particular moment to maintain momentum [16]. Additionally, a human facilitator is compromised in their ability to contribute ideas to a session and is subject to the usual foibles of inter-personal relations [15]. AI can monitor group contributions and through reference to brainstorming rules and parameters, suggest shifts of topics, outline timings, prompt contributions, etc. The task of a workshop moderator is multi-fold. A design team moderator must listen, observe, assess, point, and direct, talk and interact with the team. In a design meeting, the moderator is the director and choreograph, allocator of roles, timekeeper and record keeper, pacemaker, and even sometimes conflict resolver. The moderator may bring in examples and own expertise to stimulate the discussion and keep the discussions focused and on track. Moderators may even introduce their own style to align the design solutions with the style and brand of the house [16]. 

Today this task is often taken by a member of the design team or a senior member of the design company [15]. Bringing all members of a design team without a human moderator into the same “room” even though the members may be physically worlds apart is the target of this project. This requires overcoming the challenging task of creating a virtual moderator that takes over the complex role of a human moderator. Team moderation was not strongly evident in current state-of-the-art (as described in Section 2) involving creative virtual environment which motivated further this exploration of the effect of moderation on the creative process through an AI agent. The attributes of the human moderator are mimicked as realistically as possible. The AI agent is represented as an avatar to enable dynamic interaction between members and assist group ideation [16]. Introducing the AI agent in the virtual environment has the advantage that meetings can be held anytime and anywhere, whilst keeping the quality of the meetings a constant high. The virtual moderator also maximises the efficiency of the team through avoiding the sacrifice of a team member for the role of the moderator. This system supports positive group dynamics, promotes unbiased and equal input from participants and prevents social loafing [22], therefore offering multiple advantages over current brainstorming scenarios. 

The virtual moderator has a positive influence on users’ behaviours and cognition, and emotions through the Proteus and social identification effects enabled by VR technology [7,53,54]. From an AI perspective, deep learning models enable empathic conversations by introducing features such as emotion recognition, eye gaze, and voice stress [73], which are crucial to building intimacy between digital humans and users [39]. Therefore, AI provides a step change towards producing enhanced methods of communication that are as realistic as possible within a virtual environment. This would potentially further enhance social identification effects and Proteus effects enabled by VR. The result entails improved immersion effects, better emotional engagement, and enhanced group ideation.

#### 6.1.2. Key Area of Intervention: Stimulation

Secondly, in relation to stimulation, participants are limited by their understanding of the subject in question. Through AI technology, an AI agent instantly accesses and presents technical knowledge to build on ideas or suggest alternatives [16]. Such knowledge includes visual stimuli through intelligent background searching and information extraction [62] such as those related to the current market and commercial information. Allowing the AI agent to present creative stimuli promotes cognitive priming effects on participants, hence encouraging ideation that is paradigm-modifying, which lead to novel solutions as opposed to leveraging of existing solutions. This addresses challenges faced during face-to-face ideation [23,26] and allows informed strategy decisions to flag up infringements or gaps relevant to the ideas in question. In addition, the AI agent assists in the classification and organisation of stimuli such as product images, 3D assets and patents, based on the session’s requirements [16]. Previous or similar designs and concepts, described by the term appraisal [9], are accessible to all members and are successfully used in the “Tech Box” concept developed at IDEO [28]. This encourages effective divergent or convergent thinking within a team, as it can ascertain what would be beneficial at a particular point in time. 

AI has brought technological advancements in the field of speech recognition [65] through speech-to-text capabilities implemented via deep learning models [66] and automatic speech summarisation techniques that offer the opportunity to extract vital information from spontaneous conversations whilst removing redundant data [68]. Automatic recording of the ideation process and products allows members to interact with previous products created and improve upon them through iterations with the aim of improving its quality. The recording process also supports multiple communication methods to foster the creative process [48,49]. Such communication methods are not limited but just verbal or written communication [54] but also include graphical and visual communication. Members can also communicate through gestures and body movement using VR tracking [14]. Additionally, they have the opportunity to choose the preferred communication method for them which motivates discussion and, hence the creativity behind concepts [7]. Speech-to-text algorithms and keyword processing generate synchonised mind maps [13] whilst sketching functionality recognises and translates free-form sketches through an AI recognition system, which facilitates interpretation [11,12]. Such functionalities directly address the limitation present in face-to-face brainstorming of production blocking [23], by limiting delays between the generation of ideas and their expression. Participants who feel less confident in sketching feel empowered to apply this medium through the facilitation and recognition capabilities offered by this system, which presents a core advantage over current brainstorming scenarios. Therefore, an AI-facilitated VR environment encourages stimulation by engaging multiple senses through the flexible application of several communication techniques, according to the individuals’ preferences. Increasing the efficiency of communication and interpretation of ideas addresses not only the limitation of production blocking, but also saves time, which makes the ideation process more cost effective and productive. Team members also have the option to maintain anonymity, therefore encouraging discussion and expression beyond social boundaries [8] if they consider it to be beneficial for their specific scenario. This also addresses traditional brainstorming limitations of social loafing [22], evaluation apprehension, and fear of negative appraisal [23]. Therefore, AI goes a step further in transcribing the spontaneous nature of communication methods which in combination with the recording abilities enabled through VR, enhances the brainstorming activity by providing users access to information that is concise and useful.

#### 6.1.3. Key Area of Intervention: Immersion

The artificial environment in current remote meetings such as videoconferencing has a negative impact on creativity since it is easier for members to divert their attention elsewhere [5]. VR directly addresses this issue since it is advantageous in terms of immersing participants into an environment, increasing emotional engagement and encouraging presence [36]. Members can concentrate on a task with less distractions, hence producing more ideas and novel products [48,49]. VR allows members to focus on the design problem to be solved through enhanced visualisation and therefore allows them to become more present in the discussion [44]. 

The level of immersion is dependent on several virtual elements. Simulated objects within the virtual environment have a crucial impact on immersion aimed at enhancing creativity [40]. Graessler and Taplick [44] agree that multi-sensory stimulation through objects, textures and materials has a positive influence on the creative outcome of the group. This is not only achieved by the virtual environment itself that positively influences individuals’ mood [45,46] but also by realistic interactions with previous design iterations (appraisal) [9]. Therefore, the multi-sensory interaction of users such as through tactility, olfactory and auditory stimuli induces a positive effect of immersion [44], hence leading to greater focus and quality ideation [48,49]. 

Implementing movement interactions in VR has proven to be highly beneficial to enhance immersion and presence [40], emotional engagement [31] and improve focus [44,54]. Through the application of AI technology, complex movements and interactions are implemented through Interactive Machine Learning (IML). This mirrors realistic ways of communication between users and enhances interactions with virtual objects [72]. In the study, performed by Kim and Jo [77], gesture-based non-verbal communication was highly effective in immersive environments which contributed to a sense of co-presence, hence the feeling of togetherness in virtual human systems. Tracing, which captures real-time capturing of the location and movements of the person within the environment, also enhances communication [48,49]. This proves that the combination of AI and VR technologies is successful in positively transforming brainstorming scenarios by involving natural interactions between team members and between humans and virtual elements, leading to enhanced immersion. The following section combines the three areas of intervention into a framework with the aim of augmenting designers’ creative experiences.

### 6.2. The Creation of an AI-Facilitated VR-Based Environment

To address the three key aspects discussed (facilitation, stimulation, and immersion) and reclaim brainstorming as a challenging and inspiring activity where participants are fully engaged, a platform for future development of an AI-facilitated, VR-based environment is proposed to support engineering design ideation. The main elements embedded within VR and AI, respectively, that are contributing towards an AI-Enhanced Virtual Brainstorming activity, are described in further detail in Figure 5.

Following the pre-determination of the type of environment required for the specific ideation scenario, the main elements brought by VR and AI technologies are integrated to generate the resulting brainstorming environment. Such elements are described in Figure 5. VR enables the configuration of the environment itself and provides access to multiple visual stimuli. Such stimuli are not only represented by avatars [50] and 3D assets [40] but also by the visual presentation of information [60,62,63]. Another main element, brought about by VR, is interactivity between the user and the surrounding environment, which is achieved through tracking [72] and recording body movements around the virtual space [48,49]. Interactivity is also enabled by multiple sensory engagements through VR’s capabilities related to tactility and haptics that simulate real-life scenarios [40,41]. On the other hand, AI supports the same interactions with an added level of complexity, demonstrating the two-way relationship between VR and AI in this framework. Through AI’s deep learning models, information can be intelligently accessed, extracted, and organised according to the team’s requirements [60,62,63]. Communication between team members is enhanced by converting speech-to-text [66] whilst providing the designers with a choice from a range of communication methods and interactions [7]. This is enabled through verbal, written, graphical and kinesthetic ways of communication. Dynamic interactions are made feasible between users and a virtual agent acting as a moderator that can also organise and analyse the information being generated [15,16].

The result is an AI-facilitated Virtual brainstorming environment that provides a transformation of current ideation processes through three main factors: facilitation, stimulation, and immersion. Team facilitation through an intelligent agent offers a multitude of benefits including natural and dynamic interactions that motivate the discussion—which is unbiased, organised and goes beyond hierarchical and social boundaries [16]. The proteus and social identification effects play an important role in positively influence members’ behaviours, cognition, and emotions [7,53,54]. The moderator also plays an active role in providing team members access to stimulations by managing information extraction processes and guiding intelligent searching [16]. 

Visual stimulation is accessed through appraisal [9] by deriving ideas from previous design iterations in digital form. Communication is not only essential between the moderator and users but also between the users themselves. The users—represented by avatars [50]—can communicate with each other via multiple techniques using all the sensory tools that users have which are linked to the voice and body movements. Gesture-based communication has been highly effective in inducing effects of co-presence [77]. When combined with other communication enhancers including emotion recognition, eye contact [73], a positive immersive experience, is achieved, which leads to creative discussions. Intelligent HCI and deep learning approaches play a vital role in enabling multiple modes of communication [75]. The environment itself also plays an important role in immersing users through positive emotional engagement facilitated by mood-enhancing surroundings [45,46]. Users can freely move around the space, and this is tracked and traced within the virtual environment to simulate realistic scenarios and increase the effect of immersion [48,49,72]. 

The AI-supported brainstorming activity will provide a step change in current ideation processes by integrating the elements brought by the environment together with those enabled by the user and link the two through a plethora of communication techniques and enhancers. The result entails a platform where participants are guided through the brainstorming process with exposure to information, environments, and artefacts that are not achievable in a conventional setting whilst ensuring the emotional well-being of designers. 

#### Challenges Faced during Facilitation, Stimulation, and Immersion

While this configuration augments the current creative experience and keeps the user central within creative process, this may also bring certain challenges. With regard to facilitation, further research is encouraged on how the appearance of the moderator can positively create proteus and social identification affects, particularly within the design engineering field. Regarding stimulation, further research is required to further enhance communication and comprehension between members such as the effects of eye contact, facial expressions, and gestures [14,56,77]. Such developments require further technological advancements in the fields of VR and AI aimed at supporting the user experience. Immersion is enhanced when the users can move and interact with the virtual space; however, this is limited by the size of the physical space itself. Alternative navigation solutions could be explored further with the aim of maintaining creative discussion. With regard to immersion, fully immersive environments suggest enhanced creativity [48,49,55]; however, user interfaces that allow the user to interact with the environment itself were shown to improve immersion [57]. Therefore, the interplay between user interfaces and immersion effects requires further study to analyse its impact in creative ideation applications.

## 7. Conclusions

This research explores how AI and VR can augment creativity within engineering design. Although there is a strong link between VR and AI technologies in the literature, the bibliographic analysis described in Section 2 has shown that further research is needed which involves the application of both technologies for creative scenarios and for engineering design applications. Therefore, this research proposes a novel framework for an AI-facilitated, VR-based brainstorming environment to support creative ideation which facilitates competitive product innovation.

Literature was reviewed to understand the current challenges in design teams associated with videoconferencing and group ideation. Such challenges are addressed through creative enhancing techniques and methods, including brainstorming [22], that are useful to provide structure to ideation and therefore generate higher creative output [22]. However, there lies the question of how current ideation techniques can be transformed to not replace but enhance human creativity with the aid of cutting-edge technologies. Face-to-face brainstorming activities also have their limiting factors such as social loafing [22], production blocking and paradigm-preserving approaches [23]. Such limitations were directly addressed in this proposal through advanced communication functionalities, therefore increasing the efficiency of the ideation process and fostering communication. The aim of this research is therefore human-centred, which aligns with core values encompassing Industry 5.0 [10,76].

The working principles of VR and AI technologies were explored to identify how they can support the creative experience. Research related to virtual brainstorming [8] resulted in three main elements that are considered to be influential, which were related to the environment, the user and the linking element between the two—communication. AI provides a step change to these three elements. This is mainly achieved through intelligent searching and information extraction [60,62,63], and advanced forms of human interaction including body movement tracking through Interactive Machine Learning (IML) [72], speech-to-text algorithms [66] and natural language processing [63]. Combining relevant elements from both VR and AI technologies resulted in three key areas of intervention described as follows:Team facilitation which provides organised and unbiased discussion during the ideation process whilst obtaining a “feel” of the dynamics, characters, and psyche of the team members during the complex design process [15,16].Stimulation through multiple senses which positively contribute to various methods of communication, recording of information and therefore leading to creative ideation [44].Immersion achieved via advanced interaction between avatars [9], with the environment [45,46] and with digital artefacts relevant to the design problem [9,40]. This results in positive emotional engagement, enhanced social identification effects and Proteus effects enabled by VR, improved focus due to the increased immersion effects and therefore a higher quality of group ideation [7,53,54].

Table 2 describes how the three main areas that support the creative design ideation process in this research are mapped according to their respective technology, whether the area is more predominantly focused on AI or VR. It is evident that VR plays a key role towards immersing users into the environment through sensory stimulation with the aim of inspiring creative ideation. On the other hand, AI plays an essential role towards enhancing communication between group members which is further augmented through dynamic moderation.

The three key areas of facilitation, stimulation and immersion were overlayed onto the bibliographic analysis discussed in Section 2, to observe where these areas would be closely linked, as shown in Figure 6. Elements that support immersion are predominantly associated with the green keyword group (Group B), where VR-related terms and applications lie. On the other hand, stimulation and facilitation are more linked with Group C which is strongly linked to AI technology. Since facilitation is highly associated with brainstorming activities, links are generated between Group C and Group A which support the less dominant keyword cluster involving creativity. This research also links VR and AI technology with creativity which therefore strengthens the highlighted area of opportunity.

For the first time, this research reclaims brainstorming as a challenging and inspiring activity where participants are fully engaged through the creation of a novel AI-facilitated, VR-based environment to support engineering design ideation. The result will involve a platform where participants are guided through the brainstorming process with exposure to information, environments, and artefacts that would not be achievable in a conventional setting. The conclusions presented in this research provide a new framework for future research into Industry 5.0 product development styles through a human-centric AI- and VR-based environment for creative product ideation.

## Figures and Tables

**Figure 1 sensors-23-03009-f001:**
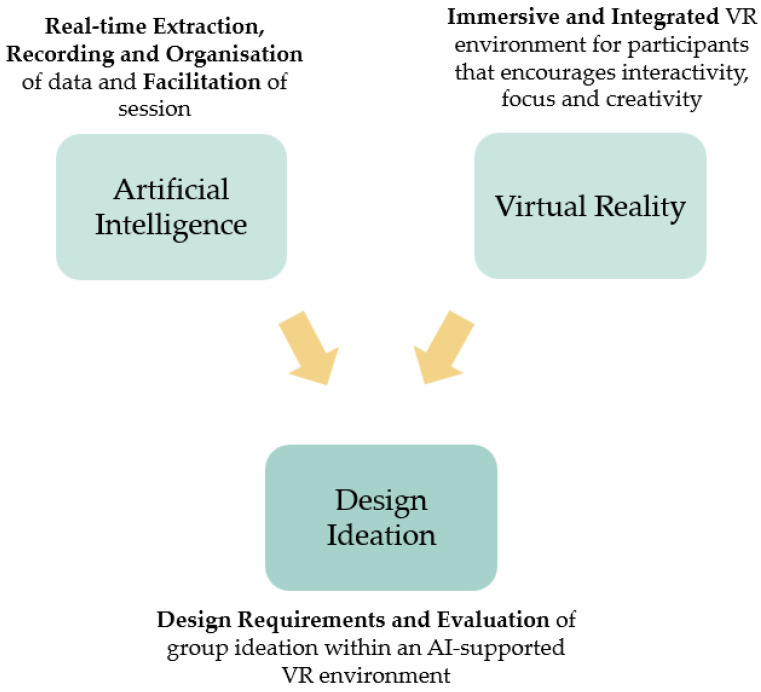
Integrating AI and VR technologies to generate a platform for augmented design ideation environments.

**Figure 2 sensors-23-03009-f002:**
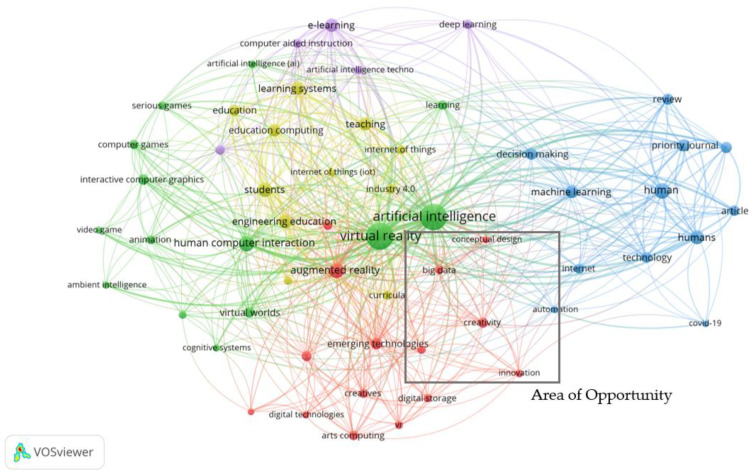
Bibliographic Representation of SCOPUS^®^ database search involving Virtual Reality (VR) technology, Artificial Intelligence (AI) and Creativity. VOSViewer^®^ (Leiden, The Netherlands) was used to produce Figure 2.

**Figure 3 sensors-23-03009-f003:**
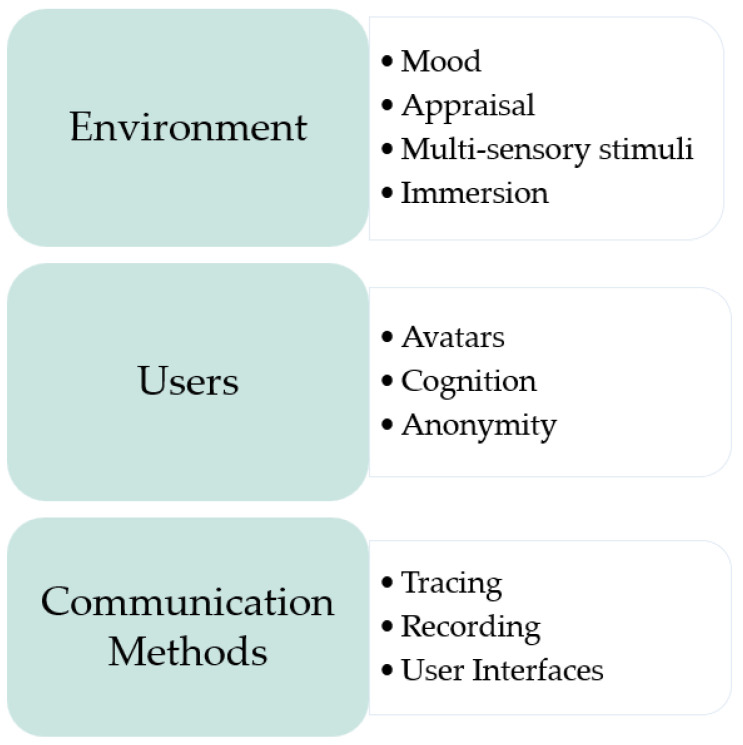
The three main elements that comprise a virtual brainstorming environment along with factors that influence their respective element.

**Figure 4 sensors-23-03009-f004:**
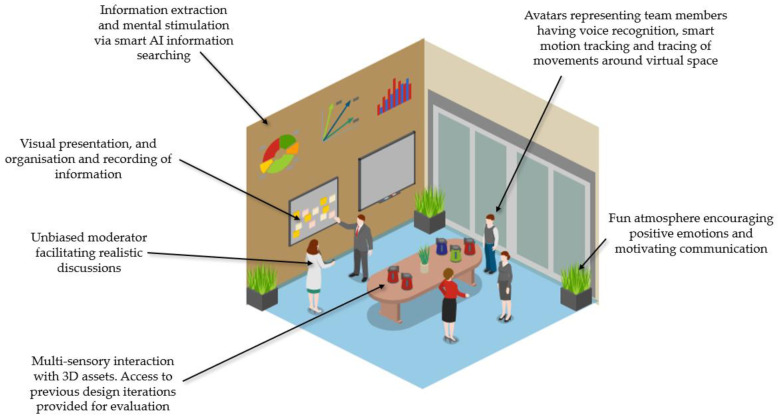
Facilitation, Stimulation, and Immersion: three key areas supporting AI- and VR-based ideation. Created using Icograms^®^ (Kyiv, Ukraine), https://icograms.com/designer (Accessed on 5 January 2023).

**Figure 5 sensors-23-03009-f005:**
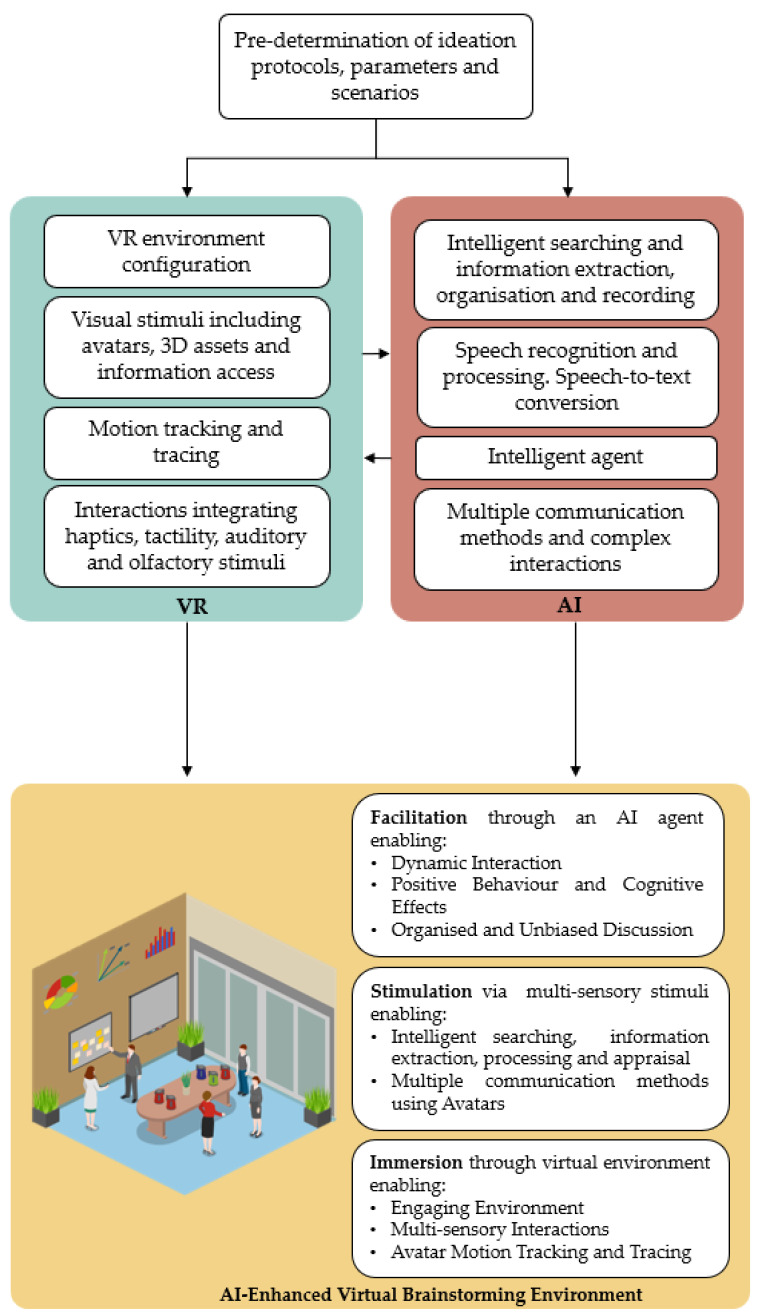
The novel AI-facilitated, VR-based Environment to Support Engineering Design Ideation. Created using Icograms^®^, https://icograms.com/designer (Accessed on 5 January 2023).

**Figure 6 sensors-23-03009-f006:**
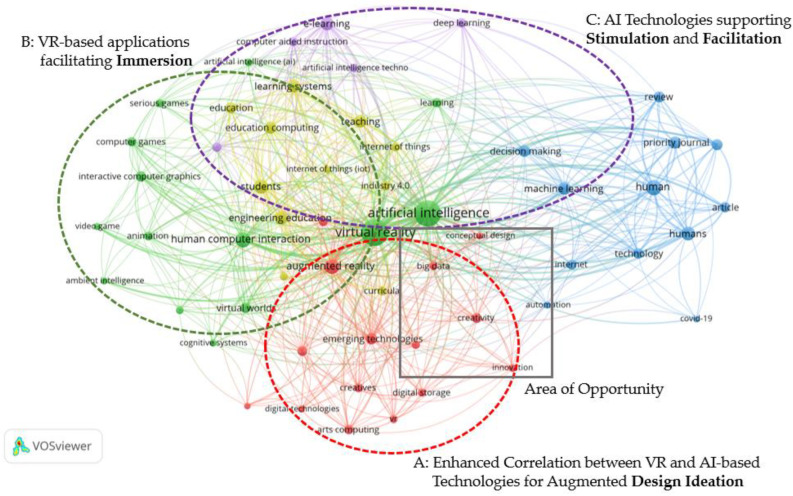
Mapping the Three Key Areas of Intervention onto the Bibliographic Representation of SCOPUS^®^ database search involving Virtual Reality (VR) technology, Artificial Intelligence (AI) and Creativity. VOSViewer^®^ was used to produce Figure 6.

**Table 1 sensors-23-03009-t001:** Commercially available collaborative ideation environments.

Product Name	Product Capabilities	Distinctive Features
Noda^®^ (Olympia, WA, USA) (Source: https://sidequestvr.com/app/1986/noda-mind-mapping-in-vr) (Accessed on 1 November 2022)	3D mind maps Storyboards Network or flow diagrams Project plans	Motion tracking Speech to text Visual iconography, which involves adding emotions and colours to concepts
MeetinVR^®^ (Valby, Denmark) (Source: https://www.meetinvr.com/2021/04/14/brainstorming-in-vr/) (Accessed on 1 November 2022)	3D mind maps Whiteboards for sticky notes, drawings, and media files Tablet based interface	Motion tracking Speech to text
Glue^®^ (Helsinki, Finland) (Source: https://glue.work/glue-platform/) (Accessed on 1 November 2022)	Presentation capabilities Whiteboards 3D model import Freehand drawings	Motion tracking AI-based facial animation, lip-syncing, and eye contact 3D spatial audio Cloud data connection
rumii^®^ (Bellevue, WA, USA) (Source: https://sidequestvr.com/app/17/rumii-for-oculus-quest-version-205) (Accessed on 1 November 2022)	Presentation capabilities Laser pointers 3D model import Screen and camera sharing Possibility to create bespoke content	Motion tracking
Arthur^®^ (Munich, Germany) (Source: https://www.arthur.digital/use-cases) (Accessed on 1 November 2022)	Pinboards Name tags associated with each user Mind mapping Breakout rooms	Sound attenuation inside breakout rooms
ShapesXR^®^ (San Mateo, CA, USA) (Source: https://www.shapesxr.com/) (Accessed on 1 November 2022)	Freeform design Storyboarding Customisable environment canvas	Motion tracking

**Table 2 sensors-23-03009-t002:** The key aspects of Facilitation, Information and Immersion are mapped according to their respective technology.

	Facilitation	Stimulation	Immersion
**AI**	Structure topics, timings and contribution based on session requirements and brainstorming best practices [16] Empathic communication through emotion recognition, eye gaze, and voice stress [73] Dynamic response and direction that is unbiased and monitored [16]	Intelligent searching and information extraction using speech-to-text algorithms [60,61,62,63] and natural language processing [63] Advanced communication techniques such as eye gaze, eye contact, voice stress [73], gestures [14], sketching [12]. Advanced interactivity through body tracking and tracing [48,49,72]	Motion tracking to enhance body language [72] and communication through complex gestures [14,77] Multi-sensory and naturalistic interaction and virtual objects [44] Empathic communication through as eye gaze, eye contact, voice stress [73] and gestures [14,77]
**VR**	Remote collaboration [7,8,9] Representation of users as avatars [50] Encouraging emotional engagement via Proteus and social identification effects [7,53,54]	Direct interaction with relevant 3D artefacts including appraisal [9] Building engagement through information presentation via moderator [16,39]	Building positive mood, focus [45,46,47,48,49] Interaction with relevant 3D artefacts including appraisal [9] User representation via avatars [50]

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
