# Peer review of "An Extended AI-Experience: Industry 5.0 in Creative Product Innovation"

_sensors, 2023, doi:10.3390/s23063009_

Round 1

Reviewer 1 Report

The manuscript reviews the leverage of VR and AI in brainstorming and proposes an AI-facilitated VR-based environment to support engineering design ideation.

The manuscript is a review article. Thus, particular information can be summarized, analyzed, and provided for readers in more detail.

1. The challenges, issues, and limitations in current (face-to-face) brainstorming activities can be elaborated in more detail. It is suggested that a table can be used to present the above information with related studies. In addition, if the performance of face-to-face brainstorming is acceptable, what is the superior benefit of the proposed AI-facilitated VR-based environment? 

2. If the quantitative or qualitative evaluations of AI-assisted brainstorming and VR brainstorming are available, please introduce, analyze, and present for readers. Thus, readers can get more insights into the necessity for the proposed AI-facilitated VR-based environment.

3. The potential challenges, issues, and limitations when leveraging the proposed AI-facilitated VR-based environment can be summarized, analyzed, and presented for readers.

Author Response

Thank you for your valuable feedback and insights. Kindly find my responses below.

Point 1: The challenges, issues, and limitations in current (face-to-face) brainstorming activities can be elaborated in more detail. It is suggested that a table can be used to present the above information with related studies. In addition, if the performance of face-to-face brainstorming is acceptable, what is the superior benefit of the proposed AI-facilitated VR-based environment? 

Response 1: Section 3.1 was updated with an elaboration of challenges, issues and limitations in face-to-face brainstorming activities. The superior benefits offered by the proposed AI-facilitated VR-based environment over current face-to-face scenarios are now explicitly reflected in Section 6.1.1 and 6.1.2 and Section 7.

Point 2: If the quantitative or qualitative evaluations of AI-assisted brainstorming and VR brainstorming are available, please introduce, analyze, and present for readers. Thus, readers can get more insights into the necessity for the proposed AI-facilitated VR-based environment.

Response 2: Section 2.1 was updated with the presentation of key outcomes of state-of-the-art studies that applied AI and VR technologies for creative scenarios that demonstrate the necessity of the proposed AI-facilitated VR-based environment in this study.  This was also reflected in Section 6.1.2.

Point 3: The potential challenges, issues, and limitations when leveraging the proposed AI-facilitated VR-based environment can be summarized, analyzed, and presented for readers.

Response 3: The new section 6.2.1 was added to explicitly discuss challenges, issues, and limitations of the proposed AI-facilitated VR-based environment.

Reviewer 2 Report

The current review aims to investigate the role of Virtual Reality and Artificial Intelligence can augment creativity within engineering design as a means to promote creative smart product innovation.

The topic is very interesting and relevant with the values of Industry 5.0 and smart product development. There are various interesting observations and conclusions. Figures and tables are useful.

I would like to make the following suggestions:

þ  Minor improvements are required in the abstract (lines 10-13) to improve coherence. For instance, you can remove duplicates (in this research, challenges).

þ  The Introduction can be strengthened. An explicit statement of questions being addressed regarding participants, interventions, comparisons, outcomes, and study design could be beneficial. It would also be useful to highlight the existing gap as well as the importance of the proposed review.

þ  It would be useful to better clarify the type of this review as well as the methodology according to which this review was conducted. Indicate if a review protocol exists. Specify study characteristics. It would be useful to make a small separate section or paragraph.

þ  It would be better if you can improve the quality of the images (ie, table 3). 

Author Response

Thank you for your valuable feedback and insights. Kindly find my responses below.

Point 1: Minor improvements are required in the abstract (lines 10-13) to improve coherence. For instance, you can remove duplicates (in this research, challenges).

Response 1: Minor improvements in abstract were performed as requested.

Point 2: The Introduction can be strengthened. An explicit statement of questions being addressed regarding participants, interventions, comparisons, outcomes, and study design could be beneficial. It would also be useful to highlight the existing gap as well as the importance of the proposed review.

Response 2: The Introduction (Section 1) was updated to explicitly state the research questions, highlight the research gaps, and demonstrate the significance of this proposal.

Point 3: It would be useful to better clarify the type of this review as well as the methodology according to which this review was conducted. Indicate if a review protocol exists. Specify study characteristics. It would be useful to make a small separate section or paragraph.

Response 3: The Research Methodology section (Section 2) was updated to specify further information including the review type and characteristics.

Point 4: It would be better if you can improve the quality of the images (ie, table 3). 

Response 4: Table 3 was updated as requested.

Round 2

Reviewer 1 Report

The issues are carefully addressed in thie revision. The efforts of the authors are highly appreciated. Because lots of modifications are made, it is suggested that the authors can proofread the manuscript again before publication.